

# Trends in summer presence of fin whales in the Western Mediterranean Sea Region: new insights from a long-term monitoring program

Paola Tepsich[1], Ilaria Schettino[2], Fabrizio Atzori[3], Marta Azzolin[4,5], Ilaria Campana[6,7], Lara Carosso[7], Simone Cominelli[8], Roberto Crosti[9], Léa David[10], Nathalie Di-Méglio[10], Francesca Frau[3], Martina Gregorietti[11], Veronica Mazzucato[7], Clara Monaco[12,13], Aurelie Moulins[1], Miriam Paraboschi[7], Giuliana Pellegrino[7,12], Massimiliano Rosso[1], Marine Roul[10], Sébastien Saintignan[5] and Antonella Arcangeli[9]

[1] CIMA Research Foundation, Savona, Italy
[2] Department of Biology, University of Padua, Padova, Italy
[3] Capo Carbonara Marine Protected Area, Villasimius, Italy
[4] Life and System Biology Department, University of Turin, Torino, Italy
[5] Gaia Research Institute Onlus, Torino, Italy
[6] Department of Ecological and Biological Sciences, Ichthyogenic Experimental Marine Center (CISMAR), Tuscia University, Tarquinia, Italy
[7] Accademia del Leviatano, Maccarese, Italy
[8] NorthernEDGE Lab, Memorial University of Newfoundland, Newfoundland, Canada
[9] ISPRA, Rome, Italy
[10] EcoOcéan Institut, Montpellier, Francia
[11] Department of Earth and Marine Science, University of Palermo, Palermo, Italy
[12] Marecamp Association, Aci Castello, Italy
[13] Department of Agriculture, Food and Environment (Di3A), University of Catania, Catania, Italy

Corresponding author
Paola Tepsich,
paola.tepsich@cimafoundation.org

## ABSTRACT

**Background.** The Mediterranean subpopulation of fin whale *Balaenoptera physalus* (Linnaeus, 1758) has recently been listed as Vulnerable by the IUCN Red List of threatened species. The species is also listed as species in need of strict protection under the Habitat Directive and is one of the indicators for the assessment of Good Environmental Status under the MSFD. Reference values on population abundance and trends are needed in order to set the threshold values and to assess the conservation status of the population.

**Methods.** Yearly summer monitoring using ferries as platform of opportunity was performed since 2008 within the framework of the FLT Med Network. Data were collected along several fixed transects crossing the Western Mediterranean basin and the Adriatic and Ionian region. Species presence, expressed by density recorded along the sampled transects, was inspected for assessing interannual variability together with group size. Generalized Additive Models were used to describe density trends over a 11 years' period (2008–2018). A spatial multi-scale approach was used to highlight intra-basin differences in species presence and distribution during the years.

**Results.** Summer presence of fin whales in the western Mediterranean area showed a strong interannual variability, characterized by the alternance of rich and poor years.

Small and large groups of fin whales were sighted only during rich years, confirming the favorable feeding condition influencing species presence. Trends highlighted by the GAM can be summarized as positive from 2008 to 2013, and slightly negative from 2014 to 2018. The sub-areas analysis showed a similar pattern, but with a more stable trend during the second period in the Pelagos Sanctuary sub-area, and a negative one in the other two sub-areas. Our findings further confirm the need for an integrated approach foreseeing both, large scale surveys and yearly monitoring at different spatial scales to correct and interpret the basin wide abundance estimates, and to correlate spatial and temporal trends with the ecological and anthropogenic drivers.

## INTRODUCTION

The fin whale *Balaenoptera physalus* (Linnaeus, 1758) is the only commonly sighted mysticete species in the Mediterranean Sea (*Notarbartolo di Sciara et al., 2003*). Genetic analyses based on both mitochondrial and nuclear DNA indicated that fin whales in the Mediterranean Sea are characterized by genetic isolation with limited but current exchange from the North Atlantic conspecific (*Bérubé et al., 1998*; *Palsbøll et al., 2004*).

The species is encountered throughout the basin, although its distribution is patchy (*Notarbartolo di Sciara et al., 2003*). Movements of the species within the Mediterranean basin do not seem to follow a clear migratory pattern, while instead the species seems to use different movement strategy, ranging between a more "traditional" latitudinal displacement to movement between specific sites characterized by patches of prey abundance, following a behavior defined as nomadic opportunistic (*Notarbartolo di Sciara et al., 2016*). A general migratory pattern, with summer concentration in higher latitudes in the north-western basin (i.e., mostly in the western Ligurian Sea and Gulf of Lion) and autumn-winter dispersal in most of the basin and towards southern latitudes, has been recently described by different studies (*Geijer, Notarbartolo di Sciara & Panigada, 2016*; *Arcangeli, Campana & Bologna, 2017*). Three main concentration areas have been identified. The first one is the Gulf of Lions and the Ligurian-Corsican-Provençal Basin, where the highest concentration of fin whales of the entire basin was recorded, especially during summer. Cetacean species concentration in the area has been the main triggering factor towards the institution here of the Pelagos Sanctuary (*Notarbartolo-di Sciara et al., 2008*). The persistence in this area is linked with high primary production (*Druon et al., 2012*), this being the only area with recurrent blooms in the entire Mediterranean basin (*D'Ortenzio & Ribera d'Alcalà, 2009*; *Mayot, 2015*) and consequently sustaining a large biomass of primary production (*Orsi Relini et al., 1998*; *Littaye et al., 2004*). Two other hotspots, coinciding with rich areas, are also known in the basin: one in the central Tyrrhenian Sea during summer (*Arcangeli et al., 2014*) and one in the Ionian Sea, around the island of Lampedusa, where a winter-spring feeding ground is reported (*Canese et al., 2006*; *Aïssi et al., 2008*). Fluctuation of local hot-spots with high inter-annual variability were related to the variability in the

pattern of productivity (*Druon et al., 2012*; *Morgado et al., 2017*), to the influence of both environmental and anthropogenic drivers of changes (*Azzellino et al., 2017*), and could be linked to biological or behavioural factors (e.g., life stage, gender, group structure) that determine small group/individual flexibility in the pattern of distribution (e.g., *Brown et al., 2016*; *Arcangeli, Campana & Bologna, 2017*).

The Mediterranean sub-population is classified as Vulnerable by the IUCN Red List of endangered species and, according to the last assessment, the population is severely fragmented; furthermore the current population trend is decreasing (*Panigada & Notarbartolo Di Sciara, 2012*). The species in the basin is facing many anthropogenic threats such as marine traffic and ship strikes (*Panigada et al., 2006*; *David, Alleaume & Guinet, 2011*; *Coomber et al., 2016*; *Peltier et al., 2019*), marine litter (*Fossi et al., 2014*; *Di-Méglio & Campana, 2017*), chemical pollution (*Marsili & Focardi, 1996*), and noise (*Sciacca et al., 2016*).

Regular systematic studies of fin whale density and abundance in the Mediterranean Basin are part of the requirements of the Marine Strategy Framework Directive (MSFD, 008/56/EC) and the Habitat Directive (HD, 92/43/EEC), but long-term basin wide information is still lacking (*Panigada et al., 2017*). Fin whale is the only species representative of the baleen whale group in the Mediterranean Sea, and consequently among the elements considered by member state for assessment of the MSFD under the Descriptor 1 - Biodiversity. Species abundance is a primary criteria (Criteria D1C2) required by MSFD for all species group for the assessment of Good Environmental Status (GES). The evaluation of trend in abundance is considered a relevant indicator to set threshold values and to express the extent to which good environmental status is being achieved (*Palialexis et al., 2019*; Art. 4 of the Decision 2017/848/EU).

Abundance and density estimates of fin whales for the Liguro-Provencal Basin (*Forcada, Notarbartolo Di Sciara & Fabbri, 1995*; *Forcada et al., 1996*), relevant to the Pelagos Sanctuary (*Gannier, 2006*) and for the Ligurian sea (*Laran et al., 2010*) from ship based surveys performed during summer, showed a strong decreasing trend, even if difficulties in comparing those estimates must be taken into account (*Panigada et al., 2011*). Latest available abundance and density estimates for this area were obtained by aerial surveys performed in 2009 and from 2010 to 2013 from Italian research groups (*Panigada et al., 2011*; *Panigada et al., 2017*) and from 2011 and 2012 from French groups (*Laran et al., 2017*). Despite the similar platforms used for the surveys, strong differences were found in the final estimates: for the Pelagos Sanctuary, as an example, 330 (95%CI [172–633]) individuals were estimated from the first groups (*Panigada et al., 2017*) and around 1100 (95% CI [600–2400]) from the second (*Laran et al., 2017*).

It has to be stressed that no abundance or density estimates exist for all the other areas of the Mediterranean sea. While *Panigada et al. (2017)* extended survey coverage to the southern areas of the western basin, no fin whale sighting was registered in those areas. Only recently the ACCOMBAMS surveys initiative (*Panigada et al., 2019*) aimed to fill this knowledge gap, with a survey that covers the entire Mediterranean Basin.

The use of ferries as platforms for conducting dedicated research has been increasing in recent years. It has been demonstrated that data collection following specific protocols can

result in the successful use of the data for species distribution studies (*Kiszka et al., 2007*; *Arcangeli, Marini & Crosti, 2012*; *Aïssi et al., 2015*; *Correia et al., 2015*; *Morgado et al., 2017*; *Azzolin et al., 2020*).

In the Mediterranean Basin, several research institutions, scientific associations and ferry companies are collaborating in the development of the Fixed Line Transect Mediterranean Monitoring Network (FLT Med Net). This project, coordinated by ISPRA, started in 2007 and it expands year by year, with the adding of new institutions and new monitored routes distributed in the central-western Mediterranean and the Adriatic and Ionian Region. A dedicated protocol is shared by all research groups, ensuring a consistent and coordinated collection of data on cetaceans, sea turtles, seabirds and human impacts, such as marine traffic and marine litter (*ISPRA, 2016*). Data collected were used for investigating species distribution, habitat preference as well as quantifying impact of human activities.

In this work we used data on fin whale distribution collected by the network operating in the central-western Mediterranean and Adriatic region. The dataset encompassed 11 years, allowing an evaluation of short-term trends (*Palialexis et al., 2019*). We investigated interannual variability in species density, presence and group size in different sub-areas. Finally, we tested the usability of the data to assess a trend, for the purposes of assessment and reporting under the Habitat Directive and MSFD.

## MATERIALS & METHODS

### Data collection

For this analysis, we used data collected during summer months (considered from late May to end of September), from 2008 to 2018. Analysis was restricted to summer months as only three of the monitored routes are covered also during winter time (as ferries either do not travel along those routes during winter or they travel during night time), consequently the winter dataset is not considered representative for the purpose of this analysis. Data were collected from dedicated observers embarked on board ferries along fixed routes, covering the western Mediterranean and Adriatic region. Sampled routes are shown in Fig. 1.

Surveys were conducted from the ferries' command deck by a team of at least three MMOs (Marine Mammals Observers) that scanned a sea area of approximately 130° from the bow abeam, to the left and right side of the command deck by naked eye and using binoculars ($7 \times 50$) to confirm species and group size. The back of the route is scanned only occasionally to avoid risk of re-counting of sightings. Considering the small but still present cross over search area from observers across the bow, sightings that could be duplicate of already registered ones where noted and not considered in the analysis. The same precautionary approach was taken for possible resightings of whales first sighted at long distance and then closer to the ferry, after dive time (set at 10–12 min taking into account both diving time of species and ferry speed). Track line of effort was recorded continuously along the survey using a dedicated Global Positioning System (GPS). Weather conditions were recorded at the beginning of the survey and every time a change occurred. Weather data included wind speed and direction, sea state (following the Beaufort scale), cloud cover,
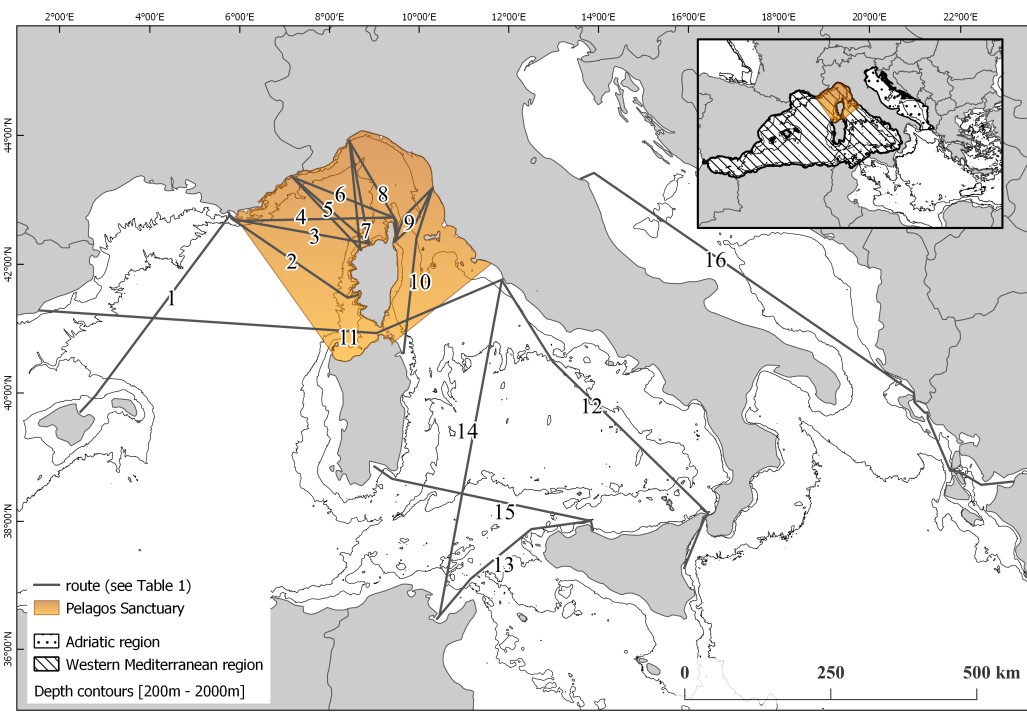

**Figure 1** **Map of the monitored routes.** (1) Toulon (FR)—Alcudia (ES). (2) Toulon (FR)—Ajaccio (FR). (3) Toulon (FR)—Ile rousse (FR). (4) Toulon (FR)—Bastia (FR). (5) Nice (FR)—Calvi/Ile Rousse (FR). (6) Nice (FR)—Bastia (FR). (7) Savona (IT)—Calvi/Ile Rousse (FR). (8) Savona (IT)—Bastia (FR). (9) Livorno (IT)—Bastia (FR). 10 Livorno (IT)—Golfo Aranci (IT). (11) Civitavecchia (IT)—Barcellona (ES). (12) Catania (IT)—Civitavecchia (IT). (13) Palermo (IT)—Tunisi (TU). (14) Tunisi (TU)—Civitavecchia (IT). (15) Cagliari (IT)—Palermo (IT). (16) Ancona (IT)—Patras (GR).

visibility and rain. Effort was considered only under optimal conditions (Beaufort equal or less than 4, good visibility). Every time a cetacean sighting occurred, the following data were recorded: time, longitude, latitude, radial distance, angle between sighted animal/group and ferry route, species, number of individuals (expressed as minimum, maximum and best estimation), behavior toward the ferry (indifferent, escaping or approaching) and any peculiar observed behavior.

Radial distance was measured using either a rangefinder stick (*Wright & Cosentino, 2015*) or a binocular with reticle rangefinder. In this latter case, distances were subsequently converted into kilometric distances applying the formula from *Kinzey & Gerrodette (2003)* (see *Cominelli et al., 2016* for more details on conversion). Angle between cetacean sighting and ferry course was measured using a compass or a protractor, set with the 0° coinciding with the bow of the ferry.

## Data preparation

All data was imported into the software QGIS and mapped using the EPSG3035 projection. GPS points of the ferry track were used to create a transect for each trip, considering a single trip from port to port. After eliminating points where weather conditions were not ideal and during which observers were not on-effort, total length of each obtained transect

**Table 1 Summary of routes (see Fig. 1 for reference), transect groups, sampled years, total number of transects monitored, number of transect discarded as not reaching the threshold value, maximum length of transects in the transect group and total km sampled along the route.**

| Route | Transect group | Years | N° transects [discarded] | Maximum—Total Length (km) |
|---|---|---|---|---|
| 1 Toulon—Alcudia | TAL | 2018 | 1 | 287.2–287.2 |
| 2 Toulon—Ajaccio | TAJ | 2011; 2014-2018 | 34 | 265.26–7,664.89 |
| 3 Toulon—Ile rousse | TI | 2018 | 2 | 159.9–294.3 |
| 4 Toulon—Bastia | TB | 2018 | 1 | 195.8–195.8 |
| 5 Nice—Calvi/Ile Rousse[a] | NC | 2009-2018 | 193 [8] | 165.71–26,409.85 |
| 6 Nice—Bastia | NB | 2017-2018 | 7 | 217.33–1,286.85 |
| 7 Savona—Calvi/IleRousse | SC | 2013-2015; 2018 | 52 | 178.01–7,954.03 |
| 8 Savona—Bastia | SB | 2008-2018 | 260 [27] | 189.32–38,127.85 |
| 9 Livorno—Bastia | N_LB | 2008; 2010-2016 | 73 | 115.03–7,874.55 |
| | S_LB | 2008-2018 | 141 [1] | 119.32–14,531.67 |
| 10 Livorno—Golfo Aranci | LGA | 2012-2018 | 110 [1] | 298.49–26,051.97 |
| 11 Civitavecchia-Barcelona | W_CVBA | 2012- | 61 [1] | 529.29–26,793.5 |
| | E_CVBA | 2018 | 62 [2] | 537.17–24,698.46 |
| 12 Catania-Civitavecchia | CTCV | 2010-2011 | 43 | 631.82–17,324.55 |
| 13 Palermo-Tunis | PATU | 2014-2018 | 27 | 349.26–6,423.87 |
| 14 Tunis-Civitavecchia | N_TUCV | 2014- | 5 | 342.42–1,337.61 |
| | S_TUCV | 2015 | 4 | 275.92–1001.96 |
| 15 Cagliari—Palermo | CAPA | 2014-2018 | 52 [1] | 396.59–13,577.55 |
| 16 Ancona—Patras | N_AP | 2015- | 11 [1] | 439.83–2,785.84 |
| | S_AP | 2017 | 9 [1] | 410.01–2,635.21 |

**Notes.**
[a]Give the proximity of the two ports of Calvi and Ile-Rousse, trips directed to either of the two ports were considered as belonging to the same transect group.

was then computed. Transects were then grouped into Transect-Groups according to the route and the sea area covered. Consequently, routes along which sampled transects differ from outbound and return, thus not sampling the exact same transect, were divided into separate transect groups (Table 1-Fig. 2). For each transect group then, the maximum length recorded for a single transect was used in order to set a threshold value for assessing transect representativeness: within each transect group, transects not reaching the 30% of the maximum length were discarded from the analysis. For each transect finally, total number of fin whale sightings and total number of individuals sighted was computed.

For the study we set two different geographic scales. The overall dataset was used to describe distribution and trend of the species at global scale, encompassing the western Mediterranean area and the Adriatic region. We then highlighted 4 sub-areas: the Pelagos Sanctuary (PEL), which includes the transect groups TI, TB, NC, NB, SC, SB, N_LB, S_LB, LGA, E_CVBA; the Western Pelagos (WP), including the transect groups TAL and W_CVBA; the South-Eastern Pelagos (SEP), defined by the transect groups CTCV, PATU, N_TUCV, S_TUCV, CAPA and the Adriatic region (AD), with the transect groups N_AP and S_AP (Fig. 2).

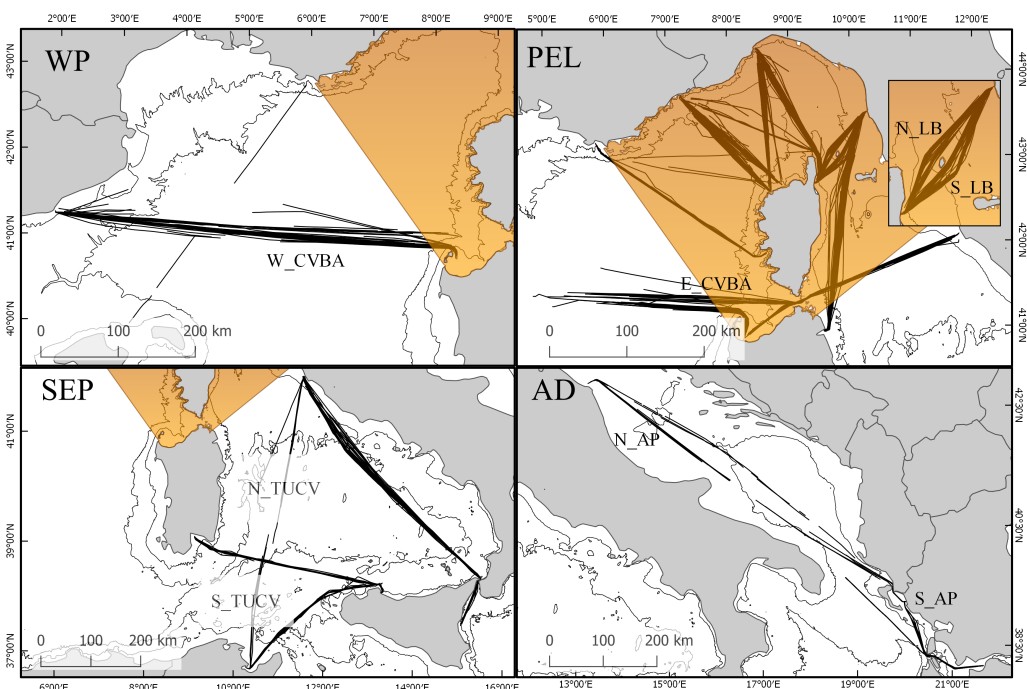

**Figure 2** **Map of sub-areas and surveyed transects.** (WP) Western Pelagos sub-area, including transects from groups TAL and W_CVBA. (PEL) the Pelagos Sanctuary sub-area, including transects from groups TI, TB, NC, NB, SC, SB, N_LB, S_LB, LGA and E_CVBA. (SEP) the South-Eastern Pelagos sub-area, including transects from groups CTCV, PATU, N_TUCV, S_TUCV and CAPA. (AD) the Adriatic sub-area including transects from groups N_AP and S_AP.

A strip-transect framework analysis was applied to the dataset. To this end, each transect was transformed into a strip-transect. In order to take into account possible differences in species detectability, distance sampling analysis has been applied. Width of each transect was then set based on the computation of the Effective Strip Width (ESW), ensuring the assumption of a 100% probability of sightings within the strip. Several factors were considered to rule out possible elements influencing detection variability among used platforms. Only data collected during sea state condition equal or less than 4 (Beaufort scale) were used in the analysis. This threshold was chosen according to results of *Cominelli et al. (2016)* where it has been demonstrated that sea state condition has no influence on sighting distance of this species from ferries, up to sea state greater than four (Beaufort scale). Another factor that has been considered is the variability of observation platforms, which directly impacts heights of observers on the sea level. Different ferries have been involved in data collection, with height of command decks ranging from 12–29 m and speed ranging from 14–29 knots. Ferries were categorized into 3 types, according to height of command deck and average speed: Type I ferries included ferries with height of command deck between 12 and 15 m and average speed 17.3 knots, Type II ferries with 20-22 m and average speed 23.1 knots command deck heights and Type III ferries with heights between 25 and 29 m and average speed 22.3 knots. Distance sampling analysis has been performed using the package RDistance (version2.1.3) in R (version 3.6.1). The objective of

the analysis was to compute the Effective Strip Width (ESW) separately for each different type of ferry used, in order to take into account differences occurring among different observation platforms (*Virgili et al., 2019*). All sightings collected during the sampling period have been used for the assessment of the ESWs. Radial distances and angles between sightings and ferry heading were used to compute perpendicular distances. For each type of ferry three different detection functions have been tested, with zero or one adjustment: Half normal, Uniform and Hazard rate. In order to choose the optimal detection function, the 6 obtained AIC have been compared and the best model has been chosen according to lowest AIC value. Finally, we tested the effect of group size as a covariate and incorporate it in the final chosen model, according to lowest AIC value.

Density of fin whales (D) was then computed as

$$D_t = \frac{n_t}{2\mathrm{ESW}_{type}l_t} * 100$$

Where

$t$ = transect

$n_t$ = number of animals observed along the transect

$\mathrm{ESW}_{type}$ = ESW as computed for the type of ferry used for that transect

$l_t$ = total length of the transect

Sampling frequency varied among routes, from weekly to monthly, depending on ferry company schedule. Spearman's rank correlation coefficient test was used in order to assess correlation among densities computed from transects of the same transect group and sampled within the same day or consecutive days, or within the same week (considered as a minimum of 7 days separating two consecutive trips). Transects were considered as correlated when Spearman's $\rho$ value was >0.5. For routes where transects were found to be correlated, one among the two consecutive transects was randomly kept. The correlation process has been done stepwise, first starting on the same day scale, then performing the analysis on the retained dataset (without transects discarded in the previous step) on the weekly scale. The same threshold for the correlation was applied to all steps.

### Fin whale groups

As school size can be an indicator of whale feeding success or food availability (*Littaye et al., 2004*), frequency distribution of size of groups was inspected at both scales (global and sub-area scale). For this analysis, group size is defined by the total number of individuals sighted at the same time after the first detection (used for the distance sampling parameters) in the area covered by the observer. Group sizes have been classified into 4 categories: "single" (for sighting with only 1 individual), "pair" (2 individuals), "small group" (3 to 5 individuals) and "large group" (more than 5 individuals).

$\chi^2$ test were used to compare frequency distribution of group sizes among years, at both scales, as well as differences in the four sub-areas.

### Summer presence

Average density for the entire basin (computed using the entire dataset of $D_t$) and for the four sub-areas (computed as the average of the $D_t$ falling within the same subarea) were

used as an index to highlight patterns in the summer presence of the species for the overall considered period. Kruskal–Wallis tests with post-hoc Dunn's tests were performed to find statistical differences among different years and among different sub-areas.

Generalized Additive Models (GAM) (*Hastie & Tibshirani, 1986*) were used to inspect the role of the year in describing the trends of the species presence, at all considered geographical scales. While linear regression methods are usually applied for inspecting trends in distribution, GAMs were preferred to linear models for their ability to deal with highly non-linear and non-monotonic relationships (*Guisan, Edwards & Hastie, 2002*), thus expected to better catch the complex trends in presence distribution known for this region (*Cominelli et al., 2016*; *Morgado et al., 2017*).

GAMs were fitted with a tweedie distribution, applying REML method for smoothness of terms and adding ESW as an offset. The only explanatory variable was the year, the scale parameter was set to −1.0 and gamma to 1.4 to deal better with overdispersion in the data (*Wood, 2006*). Final GAM formula is

$$D = s(year, k = n)$$

Knots were restricted to 7, as the shortest continuous time series available (for the WP and SEP regions, considering gap years).

## RESULTS

### ESW

During the summer months between 2008 and 2018, 228257.5 km along 1190 transects have been monitored in the Mediterranean Sea. After considering the 30% of the maximum length threshold values, 1,146 transects were kept for the analyses (Table 1).

1,705 sightings have been collected during the entire period and among these 1,687 could be used for computing ESWs. No fin whale sightings were recorded in the AD sub area, so no further analysis was possible for this sub area.

Based on AIC, for all the three groups of ferries a Hazard rate model with no adjustment terms was chosen as the final detection function. For all the three groups adding group size as a covariate resulted in a lower AIC value. More details of distance sampling analysis are reported in Supl_2_Distance.

Final ESWs were set respectively 1858 m for Type I ferries, 2,657 m for Type II e 1113 m for Type III.

### Correlation between trips

Table 2 summarizes the results of the Spearman's rank correlation coefficient for the different transect groups. Transect groups missing from the table were not tested as there were not enough occurrences for the test or there was a 0 inflation (due to a low number of sightings). Transects performed within the same day or consecutive days were correlated (Spearman's correlation index >0.5) along the SB, NC and SC transect groups, but not in the LG and CTCV groups. However, since the test is not significant for these groups, we decided to consider transects performed on the same day as correlated, following a precautionary approach. For the NLB and SLB transect groups, Spearman's correlation

**Table 2  Results from Spearmans correlation test among transects of the same group performed the same day (or consecutive days) or the same week.**

|  | df | ρ | p-value |
|---|---|---|---|
|  |  | DAY |  |
| NC | 77 | 0.59 | 7.816e−09 |
| SC | 23 | 0.69 | 0.0001413 |
| SB | 108 | 0.57 | 6.626e−11 |
| LGA | 48 | 0.18 | 0.1991 |
| CTCV | 19 | −0.15 | 0.5 |
|  |  | WEEK |  |
| TAJ | 8 | −0.45 | 0.1906 |
| NC | 66 | 0.35 | 0.003043 |
| SC | 17 | 0.49 | 0.03218 |
| SB | 93 | 0.35 | 0.0004112 |
| N_LB | 23 | 0.45 | 0.02234 |
| S_LB | 29 | −0.05 | 0.798 |
| LGA | 25 | −0.14 | 0.4836 |
| W_CVBA | 18 | 0.43 | 0.005881 |
| E_CVBA | 18 | 0.18 | 0.4498 |
| CTCV | 33 | −0.16 | 0.3522 |
| CAPA | 29 | −0.06 | 0.74 |

index is not calculated as not enough data were available, while for the NB, TAJ and TI there were only few cases of transects performed within the same day.

Weekly correlation was not found along any transect group. Following the correlation tests and not considering the N_AP and S_AP transects, 350 transects were eliminated so the final dataset consists of 796 transects.

## Fin whale Group sizes

Only sightings from transects selected after checking for correlation were used to inspect group size of the species. The final dataset accounts for 1154 sightings, for a total of 1608 fin whale sighted. More than 72% of sightings were of single individuals ($n = 834$), 21% of pairs ($n = 245$), and the remaining 7% of groups of three or more individuals ($n = 69$ and $n = 6$ for "small group" and "large groups" respectively). The main outlier is represented by a sighting of 12 individuals, occurred on 05/06/2015 along the E_CVBA transect and representing a very sparse group of animals.

$\chi^2$ test for the entire study area indicated a significant difference in group distribution among years ($\chi^2 = 66.038$, $df = 30$, $p$-value $= 0.0001613$). Overall, single individuals were the main type of encounter, followed by "pair". Small groups were detected more constantly only in 2012, 2013 and 2015 and were rare in 2009, 2010 and 2018; large groups were encountered only in 2013, 2015 and 2018.

Figure 3 represents the frequency distribution of group size by sub-areas and by years. Small and large groups of fin whales were frequently sighted only in the PEL or in the WP, in 2012, 2013 and 2015, while no groups occurred in the rest of the basin.

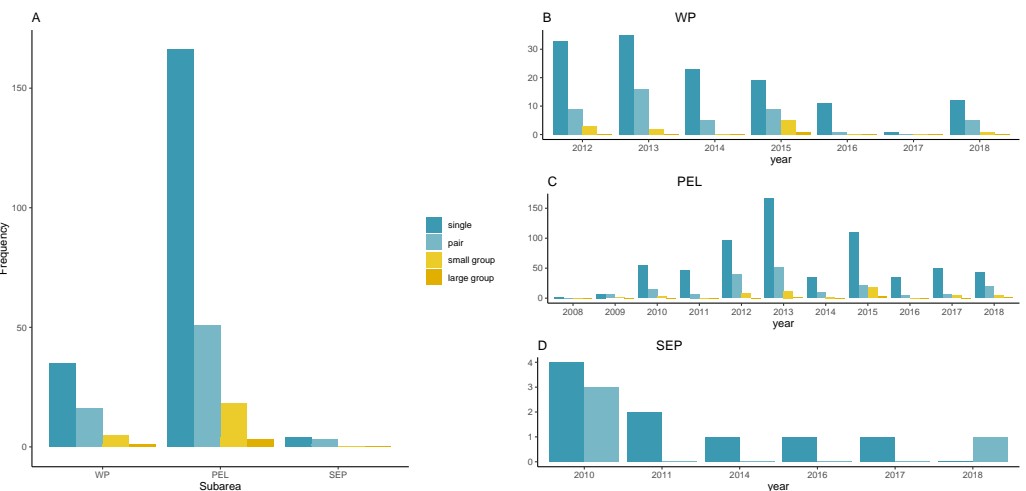

**Figure 3** **Frequency distribution of fin whale group sizes among sub-areas and per year.** (A) Frequency distribution of fin whales group sizes among sub-areas. (B) Frequency distribution of group sizes among years in the WP sub-area. (C) Frequency distribution of group sizes among years in the PEL sub-area. (D) Frequency distribution of group sizes among years in the SEP sub-area.

$\chi^2$ test found a significative difference among years in the PEL ($\chi^2 = 53.241$, $df = 30$, $p$-value $= 0.005579$), while no differences was found in the two other sub-areas ($\chi^2 = 18.128$, $df = 18$, $p$-value $= 0.4473$ for the WP and $\chi^2 = 4.9524$, $df = 5$, $p$-value $= 0.4217$ for the SEP).

## Summer presence

Overall density in the western basin for the entire period was 0.30 fin whales per 100 km$^2$ (95% CI [0.26–0.35]); the sub-area scale confirmed the importance of the PEL, where the overall D recorded was 0.35 (95%CI [0.30–0.41]). Highest D was recorded in the WP area (0.46; 95% CI [0.30–0.62]) while lowest value characterized the SEP (0.01; 95%CI [0.005–0.02]). Differences among different sub-areas were statistically significant (KW Kruskal-Wallis chi-squared $= 91.926$, $df = 2$, $p$-value $<2.2e-16$), confirmed by the Dunn's post-hoc test.

Yearly D values for the overall region and for the three sub-areas are visualized in Fig. 4 and reported in Table 3.

Kruskal-Wallis post hoc test for differences among years, statistically confirmed the interannual variability (Kruskal-Wallis chi-squared $= 74.51$, $df = 10$, $p$-value $= 5.926e-12$). Dunn's test highlighted some years as being very different from the others (Table 4). In particular 2012, 2013 and 2015, showed highest values of the considered period, with 2013 being the most anomalous year, differing from 8 other years. Looking at the poorest years, 2008 and 2014 emerge, though for 2008 it must be stressed that the low survey coverage is most probably affecting this result.

Concerning the sub-area analysis, no differences were found among years in the SEP area (Kruskal-Wallis chi-squared $= 9.1586$, $df = 6$, $p$-value $= 0.1649$), while interannual
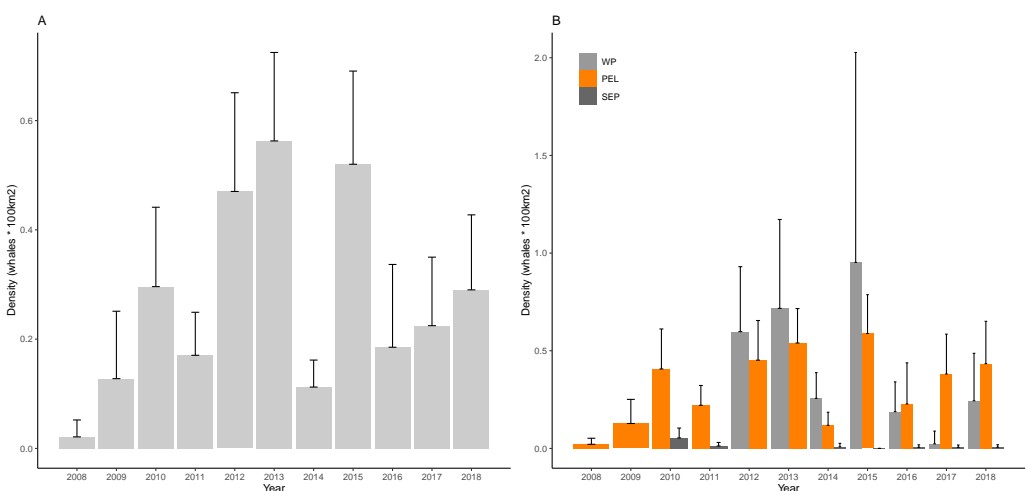

**Figure 4** **Density of fin whales in the study area and in the considered sub-areas.** (A) Mean density of fin whales per year in the Western Mediterranean basin (B) Mean density of fin whales per year in the Western Pelagos (WP), Pelagos Sanctuary (PEL) and South Eastern PEelagos (SEP) sub-areas. Error bars represent 95% Confidence Intervals.

differences were confirmed for the PEL (Kruskal-Wallis chi-squared $= 59.644$, $df = 10$, $p$-value $= 4.236e{-}09$) and for the WP area (Kruskal-Wallis chi-squared $= 17.937$, $df = 6$, $p$-value $= 0.006393$).

In PEL, the most anomalous years appeared to be 2015. Poorest years match to the ones observed for the entire region, though for 2008 we shall always consider the difference in sampling effort. Similarly, in 2013 and 2015 we find the same anomalous peaks shown for the entire basin. In this sub area 2010, 2012 and 2017 can be considered as reference years, not being different from any other year (Table 5A). For the WP area, 2015 emerges as the richest year, though not being different from any other years, while 2017 is the lowest density year and represent an anomaly compared to 2012 and 2013, as confirmed by Dunn's test (Table 5B). This sub area is characterized by high variability of D values, but the overall pattern of D is similar to the one of the PEL sub region.

Estimated trends in species presence at all considered spatial scales are shown in Fig. 5. For the Western Mediterranean basin, it is possible to summarize the trend into 3 separated periods: a positively increasing trend from 2008 up to year 2013, with the predicted density values increasing over 60% in this period, then a slightly decreasing trend up to 2016 and finally a relatively stable period during the last two considered years. A similar pattern is found in the Pelagos Sanctuary area, where after the first period with the increasing trend ending with a peak in 2012-2013, a relatively stable period is found, with a new slowly increasing trend in the end. The Western Pelagos and the South Eastern Pelagos area showed an almost linear negative trend. Gaps in the dataset for these two sub areas, as well as a lower coverage though, must be taken into account in the analysis of these trends. While for the WP and SEP sub areas, the year has a good ability to explain deviance of the dataset (deviance explained is 15.1% and 33.2% respectively for the WP and SEP sub regions,

**Table 3  Yealy fin whale Desnities for the Western Mediterranean Basin and relative sub-areas.** Yearly mean Density (D —whales * 100 km$^2$) and 95% Confidence Interval, for the (a) entire study area and for the (b) different considered sub-areas.

**(A)**

| Year | D (whales * 100 km$^2$) | 95% CI |
|---|---|---|
| 2008 | 0.02 | −0.01–0.05 |
| 2009 | 0.13 | 0.00–0.25 |
| 2010 | 0.30 | 0.15–0.44 |
| 2011 | 0.17 | 0.09–0.25 |
| 2012 | 0.47 | 0.29–0.65 |
| 2013 | 0.56 | 0.40–072 |
| 2014 | 0.11 | 0.06–0.16 |
| 2015 | 0.52 | 0.35–0.69 |
| 2016 | 0.18 | 0.03–0.34 |
| 2017 | 0.22 | 0.10–0.35 |
| 2018 | 0.29 | 0.15–0.43 |

**(B)**

| Year | D (95% CI) | | |
|---|---|---|---|
| | WP | PEL | SEP |
| 2008 | na | 0.02 (−0.01–0.05) | na |
| 2009 | na | 0.13 (0–0.25) | na |
| 2010 | na | 0.41 (0.20–0.61) | 0.05 (0–0.10) |
| 2011 | na | 0.22 (0.12–0.32) | 0.01 (−0.01–0.03) |
| 2012 | 0.60 (0.26–0.93) | 0.46 (0.25–0.65) | na |
| 2013 | 0.72 (0.26–1.17) | 0.54 (0.36–0.71) | na |
| 2014 | 0.25 (0.12–0.39) | 0.12 (0.05–0.18) | 0.01 (−0.01 0.02) |
| 2015 | 0.96 (−0.12–2.03) | 0.59 (0.39–0.79) | 0 |
| 2016 | 0.23 (0.02–0.44) | 0.23 (0.02–0.44) | 0.01 (−0.01–0.02) |
| 2017 | 0.02 (−0.04–0.09) | 0.38 (0.18–0.58) | 0.01 (−0.01–0.02) |
| 2018 | 0.24 (0–0.49) | 0.43 (0.22–0.65) | 0.01 (−0.01–0.02) |

Table 6A), lower values are found for the PEL sub region (deviance explained 2.39%). Being this subregion the most covered spatially and temporally, we tried to investigate possible factors affecting variability other than the year. We then fitted the GAM model for this sub region adding the transect_group as variables in the model. By adding the routes, the new model has a deviance explained of 44.1%. Particularly, the easternmost routes, namely the Livorno-Bastia, Livorno-Golfo Aranci e Savona-Bastia act as significatively contributing factors (Table 6B).

# DISCUSSION

The Marine Strategy Framework Directive aims to achieve Good Environmental Status (GES) of the EU's marine waters by 2020 and to protect the resources upon which marine-related economic and social activities depend. Population abundance is one of the criteria (D1C2) identified to assess progress towards the GES, and specifically baleen

**Table 4 Dunn's test results for the Western Mediterranean basin.** *P*-values of the Dunn's test are reported. Bold cells highlight significative differences among years ($\alpha$ is set to 0.05, $H_0$ is rejected if *P*-value is $>= \alpha/2$). Along the diagonal, the number of differences from other years are summarized.

| | 2008 | 2009 | 2010 | 2011 | 2012 | 2013 | 2014 | 2015 | 2016 | 2017 | 2018 |
|---|---|---|---|---|---|---|---|---|---|---|---|
| 2008 | 6 | 0.2298 | **0.0033** | 0.0519 | **0.0001** | **0.0000** | 0.0528 | **0.0000** | 0.0439 | **0.0220** | **0.0028** |
| 2009 | | 5 | **0.0080** | 0.1527 | **0.0001** | **0.0000** | 0.1565 | **0.0000** | 0.1270 | 0.0635 | **0.0067** |
| 2010 | | | 3 | 0.0375 | 0.0491 | **0.0023** | 0.0295 | 0.0376 | 0.0754 | 0.1781 | 0.4783 |
| 2011 | | | | 3 | **0.0001** | **0.0000** | 0.4791 | **0.0001** | 0.4138 | 0.2381 | 0.0313 |
| 2012 | | | | | 6 | 0.1122 | **0.0001** | 0.4546 | **0.0008** | **0.0053** | 0.0530 |
| 2013 | | | | | | 8 | **0.0000** | 0.1325 | **0.0000** | **0.0001** | **0.0025** |
| 2014 | | | | | | | 4 | **0.0000** | 0.3928 | 0.2178 | **0.0242** |
| 2015 | | | | | | | | 6 | **0.0005** | **0.0037** | 0.0406 |
| 2016 | | | | | | | | | 3 | 0.3206 | 0.0661 |
| 2017 | | | | | | | | | | 4 | 0.1626 |
| 2018 | | | | | | | | | | | 4 |

**Table 5 Dunn's test results for the PEL and WP sub-areas.** *P*-values of the Dunn's test for the (a) PEL sub-area and for the (b) WP sub-area are reported. Bold cells highlight significative differences among years ($\alpha$ is set to 0.05, $H_0$ is rejected if *P*-value is $>= \alpha/2$). Along the diagonal, the number of differences from other years are summarized.

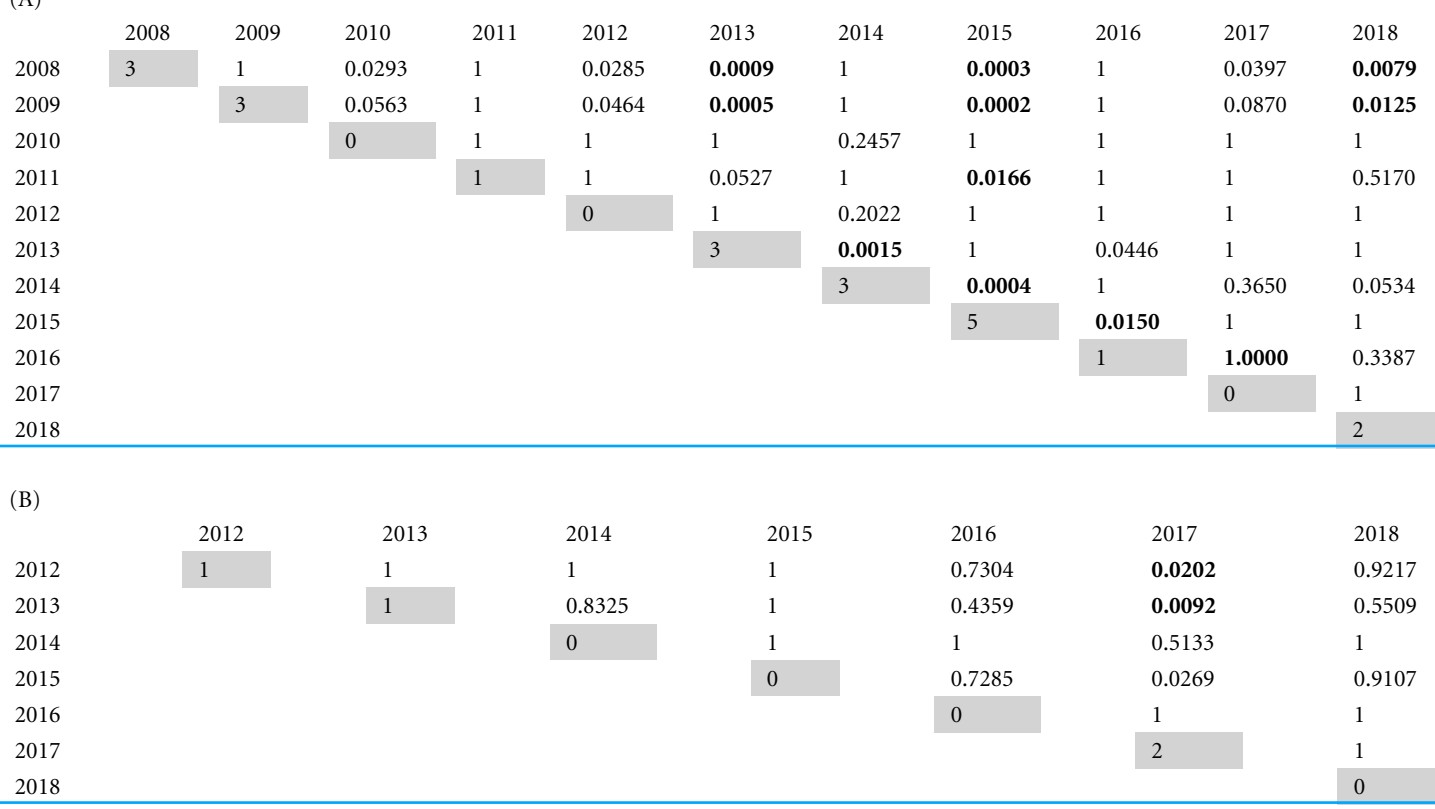

(A)

| | 2008 | 2009 | 2010 | 2011 | 2012 | 2013 | 2014 | 2015 | 2016 | 2017 | 2018 |
|---|---|---|---|---|---|---|---|---|---|---|---|
| 2008 | 3 | 1 | 0.0293 | 1 | 0.0285 | **0.0009** | 1 | **0.0003** | 1 | 0.0397 | **0.0079** |
| 2009 | | 3 | 0.0563 | 1 | 0.0464 | **0.0005** | 1 | **0.0002** | 1 | 0.0870 | **0.0125** |
| 2010 | | | 0 | 1 | 1 | 1 | 0.2457 | 1 | 1 | 1 | 1 |
| 2011 | | | | 1 | 1 | 0.0527 | 1 | **0.0166** | 1 | 1 | 0.5170 |
| 2012 | | | | | 0 | 1 | 0.2022 | 1 | 1 | 1 | 1 |
| 2013 | | | | | | 3 | **0.0015** | 1 | 0.0446 | 1 | 1 |
| 2014 | | | | | | | 3 | **0.0004** | 1 | 0.3650 | 0.0534 |
| 2015 | | | | | | | | 5 | **0.0150** | 1 | 1 |
| 2016 | | | | | | | | | 1 | **1.0000** | 0.3387 |
| 2017 | | | | | | | | | | 0 | 1 |
| 2018 | | | | | | | | | | | 2 |

(B)

| | 2012 | 2013 | 2014 | 2015 | 2016 | 2017 | 2018 |
|---|---|---|---|---|---|---|---|
| 2012 | 1 | 1 | 1 | 1 | 0.7304 | **0.0202** | 0.9217 |
| 2013 | | 1 | 0.8325 | 1 | 0.4359 | **0.0092** | 0.5509 |
| 2014 | | | 0 | 1 | 1 | 0.5133 | 1 |
| 2015 | | | | 0 | 0.7285 | 0.0269 | 0.9107 |
| 2016 | | | | | 0 | 1 | 1 |
| 2017 | | | | | | 2 | 1 |
| 2018 | | | | | | | 0 |

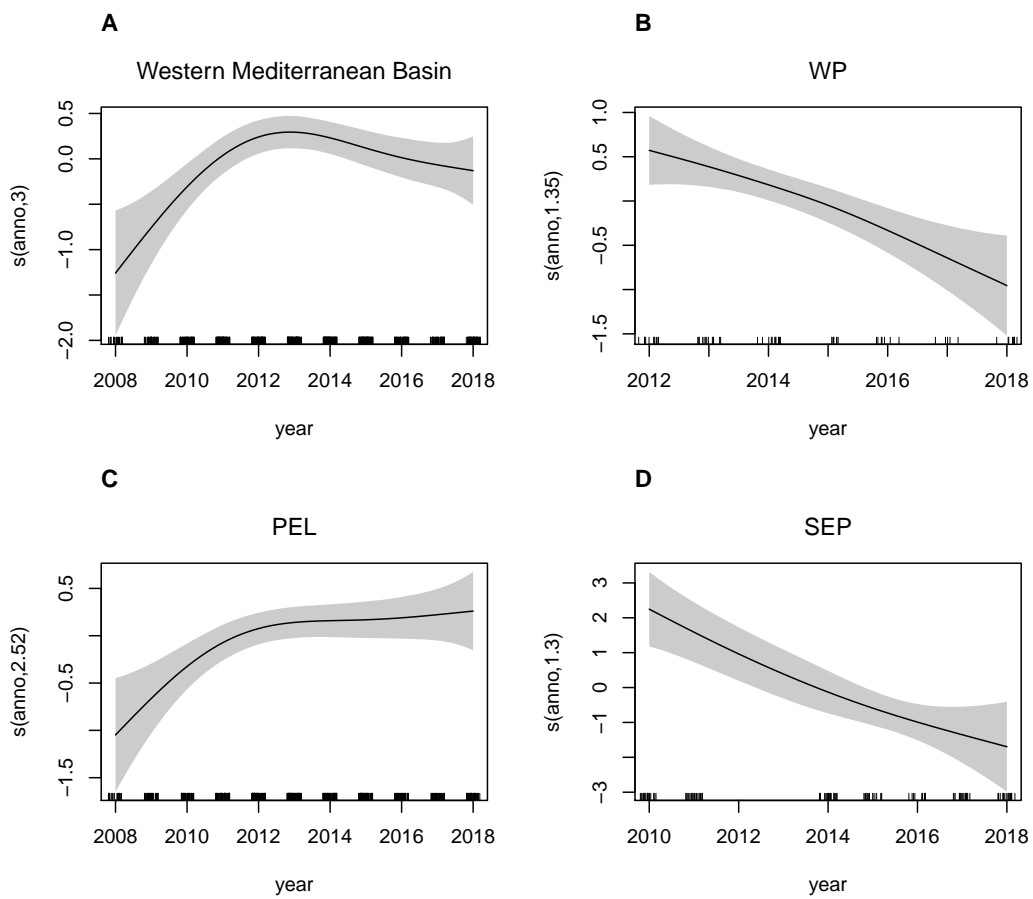

**Figure 5** **GAM plots showing fin whale density as a function of the year.** Generalized additive model (GAM) predicted smooth splines of the fin whale density as a function of the year. Tick marks above the *x*-axis indicate the distribution of observations. Shaded area represents the 95% confidence interval of the smoothspline function. This plot can be used to understand trends of speciespresence in the Western Mediterranean Basin (A) as well as for the three sub-areas: Western Pelagos (B), Pelagos Sanctuary (C) and South-Eastern Pelagos (SEP).

whale trends in abundance, is one of the criteria for assessing GES under the biodiversity descriptor D1. Monitoring of fin whale abundance is also required for the assessment of the status of the species for conservation purposes (e.g., IUCN assessment, conservation objectives under the Pelagos Sanctuary, ACCOBAMS agreement, Barcelona Convention). It is widely recognized that long dataset are needed for reliable trend estimation, and 10 years have been identified as a suitable interval for the short-term trend assessment (*Palialexis et al., 2019*). Recurrent monitoring of large areas though is difficult to achieve, for logistic and furthermore cost related factors. In this work, we present the results from a long-term monitoring project, which allowed for the creation of a 10year dataset on fin whale presence in the Western Mediterranean. Data have been collected using ferries as platform of opportunity and following a dedicated research protocol, ensuring data consistency. Effectiveness of this ferries fleet has been previously demonstrate in monitoring of marine litter (*Arcangeli et al., 2020*), sea turtles (*Arcangeli et al., 2019*) and

**Table 6  Results of the Generalized Additive Models area reported.** (a) GAMs for the Western Mediterranean Basin and for the three considered sub-areas. (b) GAM for the PEL sub-area, adding the transect group as an explanatory variable.

| | Estimate | edf | SE | t | F | P |
|---|---|---|---|---|---|---|
| **(A)** | | | | | | |
| Western Mediterranean Basin | | | | | | |
| *Intercept* | −3.03679 | | 0.07066 | −42.98 | | <2e-16*** |
| year | | 2.998 | | | 5.037 | 0.000964 *** |
| Deviance explained | | 2.61% | | | | |
| WP | | | | | | |
| *Intercept* | −2.0225 | | 0.1348 | −15.01 | | <2e-16*** |
| year | | 1.481 | | | 7.14 | 0.00312 ** |
| Deviance explained | | 15.1% | | | | |
| PEL | | | | | | |
| *Intercept* | −2.97249 | | 0.07683 | −38.69 | | <2e-16*** |
| year | | 2.524 | | | 4.415 | 0.00462 ** |
| Deviance explained | | 2.39% | | | | |
| SEP | | | | | | |
| *Intercept* | −6.5430 | | 0.4215 | −15.52 | | <2e-16*** |
| year | | 1.297 | | | 13.58 | 0.000449 *** |
| Deviance explained | | 33.2% | | | | |
| | | | | | | |
| **(B)** | | | | | | |
| PEL | | | | | | |
| Intercept | −2.3663 | | 0.2034 | −11.585 | | <2e-16 *** |
| TI | −57.0810 | | 68080.9145 | −0.001 | | 0.999 |
| TB | −2.9701 | | 1.196 | −1.352 | | 0.177 |
| NC | 0.1646 | | 0.2274 | 0.724 | | 0.469 |
| NB | −0.9905 | | 0.6048 | −1.638 | | 0.102 |
| SC | 0.4763 | | 0.2938 | 1.621 | | 0.105 |
| SB | −1.1578 | | 0.2457 | −4.712 | | 3.06e-06 *** |
| LB | −3.9503 | | 0.4001 | −9.873 | | <2e-16 *** |
| LGA | −3.8395 | | 0.4444 | −8.639 | | <2e-16 *** |
| E_CVBA | 0.2890 | | 0.2706 | 1.068 | | 0.286 |
| Smooth term | | | | | | |
| year | | 2.964 | | | 2.3333 | 0.0453 |
| Deviance explained | | 44.1% | | | | |

cetacean species (*Campana et al., 2015*; *Arcangeli, Campana & Bologna, 2017*; *Morgado et al., 2017*). In this work we further addressed possible biases arising from sampling methodologies, in several steps of the analysis framework. Ninety two percent of our dataset has been collected with sea state conditions equal or less than three (Beaufort scale); considering that a threshold of 4 can be applied for ensuring highest sightability of this species from the considered platforms (*Cominelli et al., 2016*), our dataset can be considered robust to weather conditions. Though different types of ferries have been used
in the analysis, the protocol foresees the use of a restrict range of ferry types as platform of observation, avoiding for example fast ferries or cruise ships. Ferries operating along the considered routes do not vary annually, neither does the effort or the survey protocol, so eventual bias are consistent along the years, thus not influencing the overall observed trend. To further consider possible influences related to differences in platform used, ferries have been treated separately for computing ESWs, based on both height of command deck and average speed. Considering that the analysis is species-specific and that the sea state threshold applied guarantees robustness towards this parameter, the computation of ESW specifically according to platform type allows for the direct comparison of different surveys (*Hedley & Buckland, 2004*; *Virgili et al., 2019*).

Fin whale summer presence in the Western Mediterranean basin is characterized by a strong interannual variability. The analysis of density indexes, performed thanks to a synoptic data collection over the western Mediterranean basin, suggests variability between high density (rich) years, as 2012, 2013 and 2015 and low density years, 2014 specifically, as for 2008 a lower research effort likely contribute to low density.

Looking at intra-basin presence and distribution, no sighting of the species occurred during the summertime over the considered period in the Adriatic Sea. While the species was previously sighted in the Adriatic Region, these sightings must be considered as occasional for the species (*Notarbartolo di Sciara et al., 2003*; *Lipej, Dulčić & Kryštufek, 2004*).

The Pelagos Sanctuary for marine mammals, established in 2002 in the northern area of the Western Mediterranean basin (*Notarbartolo-di Sciara et al., 2008*), is confirmed as a very important area for this species, hosting high density values during summertime. The interannual variability is also present in this sub-area, as already found in previous works (*Panigada et al., 2005*; *Cominelli et al., 2016*; *Morgado et al., 2017*). The analysis of this biggest dataset strengthens the importance of considering this variability in planning monitoring on a yearly basis. The Western Pelagos sub-area emerged as an important area for the species (*Arcangeli, Campana & Bologna, 2017*). Density values were comparable and even higher than the ones recorded in the Pelagos Sanctuary sub-area. Values recorded here also showed the highest variability. Acoustic studies and stable isotope analysis (*Castellote, Clark & Lammers, 2012a*; *Castellote, Clark & Lammers, 2012b*; *Giménez et al., 2013*), indicate the presence in the Mediterranean basin, and particularly in the area South of Spain, of another subpopulation of fin whales, the NENA subpopulation (North East North Atlantic fin whales), seasonally travelling here from the North Atlantic Ocean. Distributional range of this sub-populations is under debate, but an extension of the range of penetration of individuals from the Atlantic to the Provençal basin has been recently demonstrated (*Giménez et al., 2013*). The highest variability recorded in the Western Pelagos sub-area can then be due to the mixing of the NENA and the MED subpopulation, occurring when the NENA fin whales travel further east than their usual distribution. It also needs to be underlined that the Western Pelagos together with the Pelagos Sanctuary sub-areas are recognized as an Important Marine Mammal Area for the Mediterranean sea (*Agardy et al., 2019*).

In the Pelagos Sanctuary sub-area, density values were found to be correlated at a daily scale, but no correlation was found at a weekly scale in the whole study area. This can be interpreted by the species not being stable in the area, probably following a patchier distribution of preys all around the basin. While a clear interpretation on whale movements is not possible through our dataset, still it gives an indication on the irregular presence of animals, consequently evidencing the need to integrate datasets coming from surveys conducted on a short period of time, which may lead to an under-estimation of results. Repeated surveys along fixed transects better catch the temporal changes in distribution of the species, integrating the results obtained by more localized surveys which focus on spatial analysis of species presence and habitat preferences.

The species is not absent from the South Eastern Pelagos sub-area, where rare but still yearly regular sightings were recorded. Specifically, localized hot-spots are known to occur in this basin during particular time of the year (*Canese et al., 2006*; *Pace et al., 2019*) while, in general, the area can be seen as a traveling area among different sub-areas. The FLT Med Network is the first and only recurring monitoring of this sub-area. Recurrent monitoring in the sub-area could help integrating short-term surveys results, especially when the fail to record species presence due to limited effort in space and time.

Looking at interannual variability at sub-area scale, some years emerge as particularly anomalous both in the Pelagos Sanctuary and in the Western Pelagos sub-area, though, in the latter, the smaller size of the dataset must be taken into account for carefully interpreting results. Still, it should be underlined that in 2012, 2013 and 2015 density values were higher than average in both sub-areas. These results confirm the pattern highlighted by *Morgado et al. (2017)*, though a strong difference in the two analyses for the year 2013 is found. In our analysis, 2013 emerges as the second richest year of the entire dataset (Fig. 4), while was classified as a poorer year than 2012 in the previous analysis. This difference can be due to the lack of data from the Tyrrhenian area, covered in this work by the transects LGA and E_CVBA, which were not considered in the previous study. The intermittent blooming area of the Bonifacio Gyre (*D'Ortenzio & Ribera d'Alcalà, 2009*) can represent an alternative feeding ground for the species that can concentrate also here, rather than in the usual areas in the Western portion of the basin (*Arcangeli et al., 2014*). This result stresses the importance of a complete coverage of the basin when looking for trends of species.

While the species is most commonly sighted as single individuals or pairs (*Notarbartolo di Sciara et al., 2003*; *Arcangeli, Marini & Crosti, 2012*), and results from this work), particularly favorable ecosystem conditions, leading to the presence of food patches, can lead to the presence of groups (*Littaye et al., 2004*; *Aïssi et al., 2008*). In our analysis, richest years, indicated by the highest density values, are also characterized by the presence of small groups and large groups in the Pelagos Sanctuary and Western Pelagos sub-areas. Only single animals were sighted on the contrary during poor years. The lack of groups in the South Eastern Pelagos sub-area seems to confirm the importance of this region mostly as a travelling area rather than a feeding ground.

While the strong interannual variability makes it difficult to highlight a linear trend, GAMs allowed for the identification of more complex trends at the basin as well as at sub-areas scales, highlighting the presence of strong peaks as well as poor years in density

values of the species. Such complex trends are likely linked to the variability of ecosystem productivity in the Mediterranean Sea (*Druon et al., 2012*; *Morgado et al., 2017*), as well as to the interrelated effect of prey availability and the impact of human pressures (*Azzellino et al., 2017*). It is interesting to note that this high variability was detected even using a dataset collected during the summer season only, which is supposed to be the season when whales concentrate mostly in the north-western Mediterranean Sea, so in the core area of the present study. While the limited survey effort in some part of the basin, does not allow for an effective abundance estimation, our dataset is catching an intra-basin variability, which can result from a redistribution of the species. The differences about rich or poor years found by our study reflect the different results in abundance estimates for the Pelagos Sanctuary obtained by *Panigada et al. (2017)* and *Laran et al. (2017)*. Indeed, for the second assessment, surveys were performed in a peaking year (2012), while the aerial surveys dataset from 2009 and 2010 were used by *Panigada et al. (2017)*. Considering that, on the basis of our findings, also the results of the ASI of 2018 could be "corrected" or at least the interpretation could be smoothed. These findings further sustain the need for a large-scale continuous monitoring in order to be able to detect the interannual component of the variability, as well as for correlating the abundance and distribution of animals with the environmental and anthropogenic drivers.

## CONCLUSIONS

The FLT Med Network, operating since 2008 in the Western Mediterranean Basin and in the Adriatic and Ionian region, is the only recurring monitoring occurring in the basin. The use of ferries as a platform of opportunity and a strong scientific protocol shared among all institutions, allow for a consistent data collection. Repeatability of surveys as well as the possibility of surveying areas usually difficult to reach, allowed for the collection of a unique dataset during the entire year. Moreover, the network is now expanding in Spain with new routes covering the recently established SPAMI Spanish corridor and in the Gibraltar Strait, and further in the southern Mediterranean basin, allowing to include already known important areas such as the Lampedusa and Malta areas.

The importance of datasets collected by platforms of opportunity has already been recognized within the MSFD and specifically for the floating marine litter monitoring in high sea areas (*Arcangeli et al., 2020*) and more recently for the sea turtles' assessment. The yearly monitoring and the GAM approach for the definition of trends, allow for the interpretation of these results within the framework of the MSFD and HD. Looking at the complex trends, we can distinguish within our sampling periods the two reference periods indicated by the HD, namely 2007–2012 as the first reporting period and 2013–2018 as the second reporting period. Keeping the spatial scale addressed by the MFSD, equal to the Western Mediterranean Basin, it is possible to confirm an increasing trend followed by a negative trend, with a −40% percentage variation from 2012 to 2018. On the other hand, the interannual analysis allowed to highlight reference years that can be used as a baseline for the definition of the trend of the following years. This is another approach that has been suggested for the evaluation of trends of population presence, given the lack of abundance

estimates in past years and the difficulties in conducting large scale surveys. We highlight 2010 and 2016 as reference years for the evaluation of the following years, being those years the less different from the others. Looking at these reference years, after a variation −43% (from 2010 to 2011), matching with the negative trend previously highlighted, the overall variation for the entire period is −6%, indicating a limited negative trend for this area.

Our results also highlighted the importance of considering different spatial scales when looking at species presence and distribution, together with the need to specifically address peculiar areas known to be important for the species.

An integrated approach foreseeing both large basin wide scale surveys and yearly monitoring, would allow a better interpretation of results. Indeed, the large basin wide scale surveys conducted every 6 or 10 years would allow for more accurate abundance estimates over the whole range of the species, while the results from yearly monitoring with ferries could help correct and interpret the large scale surveys, adding the information on interannual variability, and helping in addressing abundance estimates into rich or poor years. Our work not only confirmed some previous findings about species presence in the area but also enlarged current knowledge of species presence in other areas previously poorly investigated.

## ACKNOWLEDGEMENTS

We wish to thank all the ferry companies who are thoughtfully collaborating with us in the FLT Med Network, and specifically the Corsica Sardinia Ferries, Grimaldi Lines, Minoan and Tirrenia. We are particularly thankful to Cristina Pizzutti (CSF), Rosa Cappuccio (Grimaldi) and all the staff of the ferries for the constant and kind support on all logistic aspects. This work would not be possible without all the students, volunteers and researchers who professionally and passionately collect data onboard.

### Funding

The project was supported in 2011 by the French part of the Pelagos Sanctuary (Convention 11-011-83400), for the monitoring along routes in Pelagos. The project was supported in 2013 by ACCOBAMS, for the monitoring along routes between Italy and Tunisia. The funders had no role in study design, data collection and analysis, decision to publish, or preparation of the manuscript.

### Grant Disclosures

The following grant information was disclosed by the authors:
French part of the Pelagos Sanctuary: 11-011-83400.
ACCOBAMS.

### Competing Interests

The authors declare there are no competing interests.

## Author Contributions

- Paola Tepsich conceived and designed the experiments, performed the experiments, analyzed the data, prepared figures and/or tables, authored or reviewed drafts of the paper, and approved the final draft.
- Ilaria Schettino analyzed the data, prepared figures and/or tables, authored or reviewed drafts of the paper, and approved the final draft.
- Fabrizio Atzori, Marta Azzolin, Ilaria Campana, Lara Carosso, Nathalie Di-Méglio, Francesca Frau, Martina Gregorietti, Veronica Mazzucato, Clara Monaco, Miriam Paraboschi, Giuliana Pellegrino, Massimiliano Rosso, Marine Roul and Sébastien Saintignan performed the experiments, authored or reviewed drafts of the paper, and approved the final draft.
- Simone Cominelli analyzed the data, prepared figures and/or tables, authored or reviewed drafts of the paper, and approved the final draft.
- Roberto Crosti conceived and designed the experiments, authored or reviewed drafts of the paper, and approved the final draft.
- Léa David conceived and designed the experiments, performed the experiments, authored or reviewed drafts of the paper, and approved the final draft.
- Aurelie Moulins conceived and designed the experiments, performed the experiments, analyzed the data, authored or reviewed drafts of the paper, and approved the final draft.
- Antonella Arcangeli conceived and designed the experiments, performed the experiments, authored or reviewed drafts of the paper, and approved the final draft.

## Data Availability

Raw data is available in the Supplemental Files. Shape files are available at Zenodo:

Tepsich, Paola, Schettino, Ilaria, Atzori, Fabrizio, Azzolin, Marta, Campana, Ilaria, Cominelli, Simone, …Arcangeli, Antonella. (2020). Effort data from the FLT Mediterranean Network 2008-2018 [Data set]. Zenodo. http://doi.org/10.5281/zenodo.4120406.

## Supplemental Information

Supplemental information for this article can be found online at http://dx.doi.org/10.7717/peerj.10544#supplemental-information.

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
