# Peer review of "Trends in summer presence of fin whales in the Western Mediterranean Sea Region: new insights from a long-term monitoring program"

_PeerJ, doi:10.7717/peerj.10544_

## Round 0.1 · original submission · Major Revisions

We have now received two reviews of your manuscript from experts in the field. Both reviewers find that your dataset of ferry-based observations of fin whales is valuable/of merit, but they differ in their appraisal of your analysis and the extent of the changes necessary before the manuscript can be further considered for publication. Based on their reviews and my own reading of your work, I am afraid that your manuscript cannot be accepted in its present form.

Both reviewers agree in the need to better address important spatio-temporal limitation of your dataset (all observations were obtained in summer and in a few ferry lines) . If none of the ancillary environmental data requested by Reviewer #1 are available, the validity of your study might be seriously compromised.. Although I am inclined to think that all Pearson correlation values presented in your manuscript were significant (if otherwise, the conclusions might be severely altered), the corresponding p values need to be provided. Most importantly, you will need to critically assess the validity of your current analysis (and eventually provide an alternative one) for estimating fin whales density. Reviewer #2 explicitly asks you to reconsider changing the title based on the variable really estimated. Although the changes requested by Reviewer #2 might initially seem of lower magnitude, s/he concurs with Reviewer #1 in the need for a better, more informed explanation of the methodology used. In addition to his/her detailed list of comments, Reviewer #2 also provides an annotated manuscript that will help improve English grammar and usage.

I therefore invite you to prepare a thoroughly revised version of your manuscript detailing how you have addressed each comment by the reviewers, on a point-per-point basis. Please be aware that your revision will most likely be sent out to the same reviewers before reaching a final decision.

Reviewer 1 ·

Basic reporting

The authors present an extensive dataset on fin whales occurrence in the Western Mediterranean Sea collected from from opportunistic platforms (ferries) between 2008 and 2018. Overall, this is clearly an important effort and that there are gaps in our knowledge of how fin whales are distributed there. My feeling is that, although this is an interesting study before to be published in a peer-review journal I would suggest to correct some important issues (see below).

A short temporal variability (data collected during summer months) is a limitation for this study and more in several regions characterized by an important environmental seasonal changes. Therefore, fin whales will be present in these areas during different periods of the year, when they undertake seasonal feeding migrations into the area. It is for this reason, that authors should include environmental data in the models (SST, Chla, depth, slope, etc.). For example, information on the levels of chlorophyll-a at the date of observations and in previous dates (15 days before, one month before...). This information could help to explain the observed fine-scale differences in whales habitat use.

Experimental design

The statistical analysis, as described, is not adequate. The way the analysis is done here, the authors created one GAM in which they only included "year" as covariate, does not explain the spatio-temporal distribution of the species. Authors should include more variables within the GAMs, describe data exploration protocols to identify outliers, data variability, and study the relationships between covariates and response variable. For example, effort, zone, type of ferry, wind speed (and associated sea state) is largely related to sampling visibility and this variability could affect the observed differences between species. Why authors did not include environmental variables that could influence the presence of whales and the capacity to observe the cetaceans? Why authors did not include latitude and longitude, depth, sea state, etc as potential predictors of cetacean presence? I do not understand why authors create GAMs with only one predictor.

Another red flag of the manuscript is the the choice of statistical tests. In line 215 authors explain the use of Pearson's correlations but they did not carry out normality tests. Moreover, authors considered as correlated when Pearson's r value was > 0.5 without carrying out any kind of statistical test indicating a p value.

Validity of the findings

Authors need to further acknowledge and discuss the limitations of distance sampling techniques to estimate density (which are only mentioned in passing) and that, at times, their data and results are oversold.

The methods will need to be re-evaluated. As it is, I am not comfortable providing additional comments regarding results and discussion as these could be biased by the non-adequate analysis.

Additional comments

Suggested references to facilitate the changes suggested in the ms:

Zuur AF, Ieno EN, Elphick CS (2010) A protocol for data exploration to avoid common statistical problems. Methods Ecol Evol 1:3–14

Hastie TJ, Tibshirani RJ (1990) Generalized additive models. Chapman and Hall, London

Díaz López, B., Methion, S. (2019). Habitat drivers of endangered rorqual whales in a highly impacted upwelling region. Ecological Indicators, 103: 610 – 616.

Virgili, A., Racine, M., Authier, M., Monestiez, P., Ridoux, V., 2017. Comparison of habitat models for scarcely detected species. Ecol. Model. 346, 88–98.

·

Basic reporting

The authors have undertaken some good work to summarise a long-term data set collected along ferry routes. Such long-term datasets are often lacking regionally and the use of ferries does facilitate collection of these data in a cost-effective way. It’s an approach that has gained considerable momentum over the last decade and it’s very positive to see this paper reporting on the valid use of these data. So I can congratulate the authors for working with these data and presenting some useful results.

The structure of the paper is generally fine although I would reorder slightly to make very clear the spatial scale of the analysis. And I suggest the authors reconsider the title of the article and some headers to remove reference to abundance – this was not calculated; it is simply density. The background information appears complete and appropriately referenced. The figures are well presented although some suggestions in the “general comments” below to help clarify the spatial units on maps and also Table 1. The analyses and results do address the research questions set out in the introduction.

The “raw data” is shared but the files appear to be incomplete. As they currently stand, these data could not be used to reproduce the analysis. For example, in the sightings file the perpendicular distance has not been included. In the effort data, there are no covariates; those used in the analysis, such as sea state should be available. A metadata file or “READ ME” tab in the datasheets would be useful to describe each field and its units. The field “site ID” (sightings) and “group” (effort) need explaining and it is not clear how one could assign sightings to the transects.

The context of the paper is for assessment and reporting under MSFD on Good Environmental Status. However, the authors need to carefully review the wording related to MSFD in the manuscript as it has been misinterpreted and inaccurate in many places.

The English throughout needs to be reviewed; I appreciate the challenges of writing a scientific paper when English is not your first language. I have tried to make some suggestions in the commented pdf to assist with that review.

Experimental design

The authors have undertaken some good work to summarise a long-term data set collected along ferry routes Utilising such platforms is a cost-effective means of surveying and this paper demonstrates the value in the data collected in this way. It also highlights the importance of this ferry network in the absence of designed regular surveys to monitor cetacean populations. However, the language in the paper sometimes suggests competition between platform of opportunity and dedicated surveys (e.g. aerial line transects) but instead the strengths (and weaknesses) of the different approaches should be acknowledged. The results from these different approaches can compliment each other.

A distance sampling data collection protocol has been followed. However, some concerns are raised (see general comments) with regards to the scanning area and measurement of distance and angles. The methods also needs to mention the assumptions underpinning a distance sampling analysis and whether they believe their protocol and dataset upholds them.

The research questions are clear and the approaches taken appropriate. The description of the methods used for data analysis would benefit from some further clarification. In particular, the treatment of the data needs to be more clearly written, especially with regards to the spatial stratification of the dataset (i.e. transects into transect groups and areas). Perhaps the section on geographic scale could come earlier on in the methods section. Also needs to be consistent use of area codes (e.g. PS v PEL for Pelagos Sanctuary) throughout text and figures.

Validity of the findings

The sightings data raise some concern about how they have been collected and consequently how they have been handled for fitting detection functions (see “General Comments”). The analytical approach is appropriate, although some comments are made around the choice of method; so, some additional information to justify this would be helpful and make it clearer for the reader.

The discussion and conclusions around the key questions are generally well formulated. However, the authors need to be more precautionary when interpretating results at sub-area/basin scale when the data comes from few ferry routes (in some subareas in particular) surveyed repeatedly. Some of the shortcomings of these types of data need to be acknowledged and inference from the results not stretched. The strengths of these types of survey are in the temporal resolution and when coupled with good ferry route coverage (multiple routes spread throughout areas) can be particularly effective. In the datasets presented here, those in the Pelagos Sanctuary, best fit this description with what appears to be reasonable spatial coverage and long-term year-round data - this represents a good example of a cost-effective monitoring approach. Discussion on this could be enhanced. See general comments for how these data might be further utilised.

Additional comments

Smaller comments/edits are highlighted and commented in the uploaded pdf - this likely has to be viewed in Adobe Acrobat in order for those comments to be visible.

109-114 The wording of these sentences is not quite right and could be misleading. The Commission Decision 2017 (https://eur-lex.europa.eu/legal-content/EN/TXT/PDF/?uri=CELEX:32017D0848&from=EN) dictates that abundance of species and their distribution (including cetaceans) are criteria to be assessed as part of Descriptor 1 Biodiversity set out in MSFD. So "trends in species abundance" is not required by MSFD, although many Member States may choose to use as part of their national indicators with which Descriptor 1 is assessed. The evaluation of trend in abundance as an indicator would not necessarily lead to a threshold setting – arguably you would not need a threshold for this approach. However, if thresholds have not been established, trends can be used in assessments of GES; there are an allowable approach rather than one that is asked for in the legislation.

151 – 152 One of the advantages of surveying from ferries is that they offer good temporal resolution because they often travel year-round. I wonder why the authors decided not to use data outside the summer months – the “influence of seasonal variability” can be dealt with in the analysis, without excluding those data. Perhaps there were too few surveys outside summer months?

157 The Figure 1 could colour code the western Mediterranean basin area and the “ADRION” region.

160 Can more detail be given about the scan area. 130o either side is mentioned but is this from the bow (transect line) abeam? Is there cross over of search area from observers on the port/starboard across the bow? Scanning area should ensure the transect line is given more search time but careful not just to scan directly ahead. Also line 175 says that the zero of the protractor was pointing to the stern of the vessel – I assume this is meant to be the bow (front) of the vessel? If observers are facing backwards then this will violate distance sampling assumptions and bring into question the validity of the analysis and results

184 The assignment of transects routes to subareas is not at all clear. The text suggest there are 2 separate transect groups (I think is a mistake in the test), yet Table 1 lists unique transect groups for each route. Figure 2 gives the impression that there are 4 subareas. The relevant table, figure and text all need revising to make it clear how the data were stratified spatially.

185 – 186 You would aim to have as much of the transect surveyed as possible, and so you have discarded those transects where surveying was carried out on less then 30% of its length. However, this does not ensure “representative’ coverage because the 30% coverage may all occur at one end. Representative coverage would be one that had some survey effort on all parts of the transect or if the study is interested in habitat associations, would have some (representative) sampling of all the different habitats along the route.

189 The analysis described is a standard line transect distance sampling analysis and not a strip transect? For strip transect, ones surveys a predetermined “strip” (e.g. 500m wide) and the assumption is that all animals in that strip are counted. So density is very easily count/area of the strip. The authors are fitting a detection function to distance sampling to account for imperfect detection of animals off the transect line – this gives an estimate of the ESW which can then be used to estimate density. Was there any truncation of the data? The assumption with this method however is that all animals on the transect line are detected with certainty. This section needs review so there is not confusion of the reader with regards the approach taken,

200 The raw sightings data raise some concerns about the degree of rounding, when radial distances and angles have been recorded (the majority of values end in “0”). The fitted perpendicular distance histograms should be examined and shown per transect group to demonstrate this is not a problem or if it is, how this has been dealt with when fitting detection functions.

207 To estimate density along a transect, were the data pooled over months and years? So data from one route were pooled, correlated transects removed, and detection functions fitted on the remainder of the transect data to derive density on each transect? A mean density was derived for transect group?

211 in the equation, rather “number of individuals sighted along the transect line” shouldn’t it be the number of sightings x mean group size? There is variation in group size as described later in the manuscript.

215 What are the variables being tested in the Pearson’s correlation coefficient test? Assume you are testing for spatial autocorrelation between density of fin whales on the repeated transects? Perhaps you could have generated a variogram?

239 Would it have been possible to include “group size” in a multi-covariate detection function? Might have been important for ESW and perhaps some subarea differences that could have been captured?

245 this section is entitled “abundance”…But abundance has not been estimated. Abundance along the ferry route could be calculated but you would need to consider how useful that it is given the limited spatial coverage of a broader basin. Arguably, in the Pelagos Sanctuary it appears that there are multiple ferry routes through the area with reasonable spatial coverage. One might consider using ferry data for that area in a spatial model to predict abundance.

256 You have estimated density per transect and a mean density per transect group. Why was a GAM not fitted using density (or N abundance along the route – estimated using an effective search area & probability of detection)? ESW could have been used as an offset. Needs further justification for the approach taken.

319 The header refers to “abundance” but this has not been estimated

322 Perhaps easier to have table of these density results, that would include ESW too. But Figure 4 is very clearly presented.

346 – 348 Again, perhaps a table of the GAM model details would be better than text description of the number of knots etc. Summary table of the GAMs and outputs

367 Is it possible that fin whales are present but just not near enough to the ferry route? Are there data from autumn/winter/spring that have been looked at to verify the conclusion that they are “absent” . The last sentence of this paragraph provides reference to some sightings in the Adriatic so perhaps occasional or rare, rather than absent, better describes their occurrence in the Adriatic

387 The statement that interannual variability in the Western Pelagos is due to the influx of NENA fin whales is very definite but the previous line suggests they do not move further east than the Balearics. There is limited transect effort at the Balearics. The statement needs to be adjusted to reflect that it is possible that NENA fin whales are in the WP or further references given to support the statement.

401 Whilst repeated surveys along ferry routes can give an idea of distribution through time, their ability to provide robust information on distribution in space can be limited and should be interpreted with care

421 This is a very important point and the authors have here acknowledged the limitations in their analysis and the discussion/conclusions should keep this in mind.

414 Figure 2 in Morgado et al (2017) shows that 2013 was the second highest yearly mean encounter rate and so this is not different from the authors results. In fact, the GAM for PS suggest increasing density until 2012, then a slight dip to 2013 before increasing again - the pattern up to 2013 is the same in Morgado, atleast for encounter rate. The Whale Occurrence Index gives a slightly different picture but interpretation needs to take into account how representative of the PS area the fixed route surveys are in both studies

433 -434 Care when referring to aims of the MSFD – the main aim is not to “assess trends in
population abundance” The aim is “The Marine Strategy Framework Directive aims to achieve Good Environmental Status (GES) of the EU's marine waters by 2020 and to protect the resource base upon which marine-related economic and social activities depend.” The abundance of fin whales, and trends in abundance, could be useful as indicators for assessing the biodiversity descriptor D1 – and there is a criterion for abundance. Perhaps the authors could report whether the Mediterranean countries that border the area of their research, have a “fin whale or baleen whale abundance” indicator – this would add weight to the need for these studies. Or perhaps the Pelagos Sanctuary has conservation objectives that require monitoring of fin whale abundance?

438 The authors report density, over time, along transect routes. Interpretation of these results at the wider scale (e.g. basin) need to be caveated because in the western basin in particular, there is very limited spatial coverage throughout. So a lower density on the ferry route may simply represent a redistribution of animals rather than real change in wider density / abundance.

445 With my previous point in mind, I would use the wider scale aerial surveys (e.g. Panigada et al.) that have designed survey transects, as evidence to support your findings rather than the authors trying to explain Panigada’s findings.

469 Another MSFD point, please check and accurately state what the MSFD requirements are. Abundance of species is a criterion for descriptor 1 (see the 2017 commission decision) and so should be used in assessments of GES but "fin whale abundance" is not specified. Be careful to make a clear distinction between what MSFD says and perhaps what countries have established as indicaitors.

489 Properly designed large scale surveys provide a robust approach for estimating abundance. However, they tend to be done in a limited time window. Ferry surveys can not “correct” these surveys but they do add additional data on distribution around the surveyed routes and patterns over the entire year which are often not available from designed large scale surveys.

Table 2 Just include results for routes that were tested (leave out the empty rows and explain in text)

---

## Round 0.2 · Minor Revisions

Please first accept my apologies for taking longer than expected in coming back to you. I have now received the review of your revised version from Reviewer #1. You will see that there is still a strong request from him/her to incorporate changes that you have not done because you deemed them not necessary. Since there was initially a clear discrepancy between both reviewers in the level of changes necessary for considering this manuscript further, I offered you the opportunity to re-submit your manuscript.

After carefully balancing the previous reviews and this new one, my judgment is that your revised manuscript is worthy of publication in PeerJ. The trends shown in Fig. 5, though lacking the mechanistic explanation required by Reviewer #1, which I supported in my first decision letter, are an important addition to our knowledge about the fin whale status in the Mediterranean.

However, before a final decision is made, I am asking you to carefully respond and explicitly include how you have addressed the issues of possible changes in ferries type and speed, sampling effort, etc., ultimately potentially impacting whale visibility differences among years.

I do see a way forward for your manuscript but you need to clearly state how these potential sampling variability effects have been managed. Although I am aware that some of these concerns were already partially addressed in your responses to Reviewer #2, they need to be better explained in the revised version of your manuscript.

Reviewer 1 ·

Basic reporting

In my previous examination of the manuscript, I suggested that major revisions and a positive re-review are required before this paper is sufficient for publication in a peer reviewed journal and join the scholarly literature. After having reviewed the manuscript and not observed any changes in certain important points I still consider that the work is not of sufficient quality to be published.

There are a number of considerations that led me to this decision. The manuscript is missing a number of relevant methodological details which would be needed to thoroughly understand and replicate this study. The authors should be aware that the manuscript should be contingent on demonstrating that their analytical approach is robust.

All the suggestions given in my previous review were made to improve the content of the manuscript and not to modify the aim of the study. The authors should consider that in order to describe a temporal trend in the presence of a given species, the effects derived from sampling variability should be ruled out.

Specifically, the manuscript lays out a description of a trend in presence of whales but then relies on many assumptions (some unrelated to the analysis). The manuscript cannot conclusively prove the existence of a trend in presence of whales, because authors did not include in their analysis the potential influence of other explanatory factors (i.e. changes in effort, sea state conditions, type of vessels used, speed of vessels, etc.) largely related to sampling visibility and this variability could affect the observed differences between years.

Without that, I do not think that the manuscript warrants publication.

Experimental design

The manuscript is missing a number of relevant methodological details which would be needed to thoroughly understand and replicate this study. The authors should be aware that the manuscript should be contingent on demonstrating that their analytical approach is robust.

Validity of the findings

The manuscript lays out a description of a trend in presence of whales but then relies on many assumptions (some unrelated to the analysis). The manuscript cannot conclusively prove the existence of a trend in presence of whales, because authors did not include in their analysis the potential influence of other explanatory factors (i.e. changes in effort, sea state conditions, type of vessels used, speed of vessels, etc.) largely related to sampling visibility and this variability could affect the observed differences between years.

---

## Round 0.3 · Minor Revisions

I appreciate your efforts to address some of Reviewer #1’s concerns and I only miss that you should also touch briefly on them in the Discussion, not only in the Materials & Methods section. Please do so in a new revised version. I have also identified a few typos in the Tracked Changes document (e.g. use of commas instead of points in lines 225-226, the whole new sentence inserted in lines 542-543 needs to be corrected and re-written). I also urge you to perform a thorough final read in order to correct other typos and improve language use. I hope that after returning your manuscript for these minor revisions I can find it acceptable for publication in PeerJ.

---

## Round 0.4 · accepted · Accept

I am happy with the final amendments made to the manuscript.